# A Single Nonsynonymous Substitution in the RNA-Dependent RNA Polymerase of *Potato virus Y* Allows the Simultaneous Breakdown of Two Different Forms of Antiviral Resistance in *Capsicum annuum*

**DOI:** 10.3390/v15051081

**Published:** 2023-04-28

**Authors:** Benoît Moury, Thierry Michon, Vincent Simon, Alain Palloix

**Affiliations:** 1INRAE, Pathologie Végétale, 84140 Montfavet, France; vincent.simon@inrae.fr; 2UMR Biologie du Fruit et Pathologie, INRAE, Université de Bordeaux, CS 20032, 33882 Villenave d’Ornon, France; thierry.michon@inrae.fr; 3INRAE, GAFL, 84140 Montfavet, France

**Keywords:** plant resistance, resistance durability, *Potyvirus*, *Capsicum annuum*, cross-pathogenicity, virulence determinant, RNA-dependent RNA polymerase

## Abstract

The dominant *Pvr4* gene in pepper (*Capsicum annuum*) confers resistance to members of six potyvirus species, all of which belong to the *Potato virus Y* (PVY) phylogenetic group. The corresponding avirulence factor in the PVY genome is the NIb cistron (i.e., RNA-dependent RNA polymerase). Here, we describe a new source of potyvirus resistance in the Guatemalan accession *C. annuum* cv. PM949. PM949 is resistant to members of at least three potyvirus species, a subset of those controlled by *Pvr4*. The F_1_ progeny between PM949 and the susceptible cultivar Yolo Wonder was susceptible to PVY, indicating that the resistance is recessive. The segregation ratio between resistant and susceptible plants observed in the F_2_ progeny matched preferably with resistance being determined by two unlinked recessive genes independently conferring resistance to PVY. Inoculations by grafting resulted in the selection of PVY mutants breaking PM949 resistance and, less efficiently, *Pvr4*–mediated resistance. The codon substitution E_472_K in the NIb cistron of PVY, which was shown previously to be sufficient to break *Pvr4* resistance, was also sufficient to break PM949 resistance, a rare example of cross-pathogenicity effect. In contrast, the other selected NIb mutants showed specific infectivity in PM949 or *Pvr4* plants. Comparison of *Pvr4* and PM949 resistance, which share the same target in PVY, provides interesting insights into the determinants of resistance durability.

## 1. Introduction

Genetic resistance of plants to pathogens is an attractive means of controlling crop diseases because of its high efficacy, simplicity of implementation, relatively low cost, and lack of adverse effects on human health. In addition, resistance is usually highly specific, which avoids undesirable side effects on non-target organisms. The other side of this specificity is the narrow spectrum of action of most resistance systems in the face of the wide diversity of plant pathogens. The spectrum of action of a resistance can be considered at different taxonomic levels of the pathogen and is generally defined by the number of pathogen species and/or isolates within a species that are controlled by the resistance. Considered at the intraspecific level, the spectrum of action of resistance is related to the durability of resistance, i.e., the ability of the resistance to maintain its long-term efficacy despite widespread use under conditions favorable to pathogen development [1]. Poorly durable resistances generally have a narrow spectrum of action at the intraspecific level as the targeted pathogen populations can easily evolve into infectious forms in these plants, i.e., acquire “resistance-breaking” abilities [2,3]. Most resistances exploited by breeders have narrow spectra of resistance, usually restricted to members of a single or small number of pathogen species. Therefore, an important goal of plant breeding is to increase both the spectrum of action and durability of plant resistance to pathogens.

Potyviruses (members of genus *Potyvirus*, family *Potyviridae*) are a major constraint to pepper (*Capsicum* spp., family *Solanaceae*) production worldwide, affecting both yield and fruit quality [4]. Pepper potyviruses belong to three main clades [5,6]. The largest one is the *Potato virus Y* (PVY) clade, which includes 19 species, 10 of which infect solanaceous plants [7]. The other two clades, the *Tobacco etch virus* (TEV) and the *Pepper veinal mottle virus* (PVMV) clades comprise fewer species, all of which infect solanaceous plants. At least ten potyviruses frequently infect pepper crops. Several *Capsicum* spp. genes or gene combinations have been shown to confer resistance to potyviruses, but none of them cover all pepper potyviruses and their spectrum of action considered at the species level and their durability differ widely [3,8,9,10,11,12]. Here, we characterized the spectrum of resistance and durability potential of a new source of potyvirus resistance in *C. annuum*.

## 2. Materials and Methods

### 2.1. Plant and Viral Material

‘PM949’ is a Guatemalan accession of *C. annuum* that was previously characterized in pepper genetic resources as a novel source of PVY resistance [13]. The inbred line *C. annuum* cv. Yolo Wonder (YW) was used in all experiments as a potyvirus-susceptible control, and the doubled-haploid line ‘W4’, derived from the F_1_ hybrid (YW × ‘Criollo de Morelos 334’), was used as a reference for the dominant potyvirus resistance gene *Pvr4* [14]. These pepper accessions are maintained by the INRAE GAFL CRB-Lég team [15]. For genetic analyses, PM949 was crossed with YW and an F_2_ population was derived from the F_1_ hybrid. Plants of *Nicotiana tabacum* cv. Xanthi were used to obtain virus inocula.

Twelve isolates representing the nine species of potyvirus PVMV, PVY, TEV, *Pepper yellow mosaic virus* (PepYMV), *Pepper mottle virus* (PepMoV), *Chilli veinal mottle virus* (ChiVMV) *Pepper severe mosaic virus* (PepSMV), *Ecuadorian rocotto virus* (EcRV), and *Peru tomato mosaic virus* (PTV) were used to characterize the resistance of PM949 and W4 [14] (Table 1). A variant of the PVY SON41p infectious cDNA clone carrying the K_472_E mutation in NIb cistron (Nuclear inclusion b), i.e., the RNA-dependent RNA polymerase cistron (RdRp), has been described previously [11].

### 2.2. Virus Resistance Tests

Two types of virus inoculations on pepper plants were performed, direct mechanical inoculations and inoculations by grafting. Potyvirus isolates were propagated in *N. tabacum* cv. Xanthi. Inocula were prepared as described by Janzac et al. [14], and inoculations were performed on 3-week-old seedlings by manually rubbing the cotyledons and the first expanded leaf with the inoculum. For inoculations by grafting, PM949 scions were grafted onto 6-to-8-week-old YW rootstocks, and the rootstocks were mechanically inoculated with the virus two weeks after grafting. Inoculated plants were placed in an insect-proof greenhouse where the temperature varied between 18 and 25 °C and were tested for virus infection up to one or six months post inoculation (mpi) for mechanical and grafting inoculations, respectively.

Enzyme-linked immunosorbent assays (ELISAs) were performed on inoculated leaves or on uninoculated, apical leaves to test for the presence of virus infection at the local or systemic level, respectively. The presence of PepSMV, PepYMV, PTV, and EcRV was assessed using an antigen coated plate-ELISA (ACP-ELISA) with potyvirus-group antiserum (Agdia, Soisy sur Seine, France) according to the manufacturer’s instructions. Other potyviruses (PVY, PepMoV, TEV, ChiVMV, and PVMV) were detected by double-antibody-sandwich-ELISA (DAS-ELISA) with previously described polyclonal antisera [14,16]. Samples were considered positive when the absorbance values at 405 nm (A_405_) were at least three times higher than the mean value of healthy controls.

### 2.3. Partial Sequencing of the PVY Genome

Total RNAs were purified from virus-infected leaves using the TRI Reagent kit (MRC, Cincinnati, OH, USA) following the manufacturer’s instructions. Direct sequencing of reverse transcription-polymerase chain reaction (RT-PCR) fragments spanning the entire NIb cistron was performed for resistance-breaking PVY variants as described in Janzac et al. [11].

## 3. Results and Discussion

### 3.1. PM949 Exhibits Narrow Spectrum Resistance to Potyviruses of the PVY Clade

Upon mechanical inoculation with PVY, two types of phenotypic responses were observed in PM949 (Table 1). Isolate PVY-Chile1 [17], synonymous with Crystal1 [14], was able to induce local necrotic lesions (NLs) in inoculated cotyledons 10 days post inoculation (dpi), whereas isolate PVY-LYE72 and a population derived from the PVY-SON41p infectious cDNA clone [16] (accession number AJ439544) did not induce any reaction. Thirty dpi, no symptoms were visible in apical leaves, and no virus was detectable by DAS-ELISA or RT-PCR, regardless of the PVY isolate. At that time, all three PVY isolates induced mosaic symptoms and high level of viral accumulation at the systemic level in the susceptible control line YW. PVY was also inoculated by grafting onto PM949. All three PVY isolates induced similar responses in inoculated plants. Ten to 14 dpi, leaves of YW rootstocks showed mosaic symptoms, and 14 to 20 dpi chlorotic and necrotic lesions appeared in leaves of PM949 scions and PVY could be detected in these leaves by DAS-ELISA. The necrotic reactions observed in scions of plants inoculated by grafting with the different PVY isolates and in cotyledons of plants mechanically inoculated with isolate PVY-Chile1 suggest that the resistance of PM949 is related to a hypersensitivity reaction (HR) and are reminiscent of the dominant resistance gene *Pvr4* [14].

To get a broader view of the resistance spectrum of PM949, we inoculated it with isolates representing eight other potyvirus species and compared it to the reference line ‘W4’ carrying *Pvr4* (Table 1 in Janzac et al. [11]). PM949 did not show resistance to PepYMV, PepMoV, PVMV, ChiVMV, and TEV, with mosaic symptoms and positive ELISA in apical leaves. In contrast, PM949 was resistant to PepSMV and EcRV, with phenotypic responses similar to those expressed following inoculation with PVY-Chile1, i.e., NLs in inoculated cotyledons, but no symptoms and no virus detected in apical leaves. A third type of reaction was observed with PTV isolates Quito and Cuzqueño, which induced NLs in both inoculated cotyledons and apical leaves, resulting in positive ACP-ELISA. All control YW plants showed infections at the systemic level from 14 to 20 dpi, depending on the potyvirus. Overall, as with *Pvr4*, resistance of PM949 was limited to viruses in the PVY clade and was not effective against viruses in the PVMV or TEV clades (Table 1). However, the spectrum of action of PM949 resistance was narrower than that of *Pvr4* as it did not include PepYMV and PepMoV. The status of PTV is uncertain. It was able to infect PM949, but not W4 systemically, and induced HR-like necrotic reactions in both genotypes. It is possible that PTV triggers resistance in PM949 but that the level and/or timing of defense responses are insufficient to restrict cell-to-cell and systemic movement of the virus.

### 3.2. Genetic Inheritance of PM949 Resistance to PVY

The F_1_ hybrid between PM949 and the potyvirus-susceptible YW line was susceptible to PVY-SON41p, showing systemic infection in 20 of 20 inoculated plants, indicating that the resistance was recessive. In the F_2_ population, 159 of 278 plants (57%) were systemically infected, while the remaining plants were symptomless and DAS-ELISA negative. The simplest Mendelian inheritance model that fits this segregation pattern corresponds to two unlinked recessive genes, each sufficient to confer resistance (Khi^2^ test; *p*-value = 0.80). Resistance conferred by a single recessive gene or, alternatively, by the combination of two recessive genes does not fit the segregation data (Khi^2^ tests; *p*-value < 10^−5^). Although the resistance phenotypes expressed in PM949 and W4 are similar, their genetic determinants are different since W4 resistance is monogenic and dominant [18]. The major source of recessive resistance in pepper against potyviruses is the *pvr2* gene, which encodes an eIF4E (eukaryotic translation initiation factor 4E) and includes many resistance alleles corresponding to nonsynonymous substitutions [19]. The sequence of the eIF4E cDNA obtained from the pepper cultivar PM949 was determined as described in Ben Khalifa et al. [20] and revealed a nucleotide sequence identical to that of the susceptible genotype YW, indicating that the recessive resistance of PM949 to PVY cannot be attributed to the *pvr2* gene.

### 3.3. Inoculations by Grafting Allowed Selection of PVY Variants Adapted to Pvr4 or PM949 Resistance

We analyzed the durability potential of PM949 resistance to potyviruses using experimental evolution in the laboratory and compared it to *Pvr4*. For *Pvr4*, no natural resistance-breaking (RB) PVY isolates were observed in the field, despite extensive cultivation of *Pvr4*-carrying cultivars worldwide for over 25 years. Furthermore, no RB potyvirus mutants could be selected by direct mechanical inoculation of the *Pvr4*-carrying W4 genotype [14]. In contrast, inoculations by grafting where the susceptible line YW was used as rootstock for W4 scions, and the rootstocks were inoculated with the virus 10 days after grafting resulted in the selection of PVY RB mutants for 3 of the 5 PVY isolates tested [14]. Using PVY-SON41p as inoculum, 6 out of 25 grafted plants resulted in the selection of an RB mutant around five mpi (Table 2). No RB mutants of the other potyviruses (PepYMV, PepSMV, PTV-Quito, PTV-Cuzqueño, PepMoV-Texas, or EcRV; 10 grafts with each virus) could be selected. In a previous study, four of the six RB PVY populations (SON41p-G1 to SON41p-G4) carried the codon substitution K_472_E in the NIb cistron (RdRp), which was found to be sufficient for *Pvr4* breakdown by reverse genetics [11]. In the present study, NIb cistrons of the two PVY populations SON41p-G5 and SON41p-G6 were sequenced and found to carry three nonsynonymous substitutions (M_365_I + G_471_E + S_478_N and I_94_T + V_416_A + A_504_V, respectively).

Graft-inoculations were also performed as described above for W4 to test whether PM949 could also select for RB potyvirus mutants, using PVY-SON41p, PepSMV, the Quito and Cuzqueño isolates of PTV and EcRV (10 grafted plants with each virus). While small necrotic and chlorotic lesions were observed on PM949 scions inoculated with PVY-SON41p, the other viruses induced large necrotic lesions on the scions (Figure 1), which led rapidly (50 dpi) to death of the grafted plants. At one mpi, no RB mutants could be observed with any of the viruses, as evidenced by the absence of infection on the back-inoculated PM949 seedlings. At six mpi, all ten plants inoculated by grafting with PVY-SON41p showed confluent chlorotic lesions or mosaic symptoms on parts of the scions, and the presence of RB mutants was demonstrated after back-inoculation to PM949 seedlings, with 100% infection (20 of 20 inoculated plants) at 21 dpi (Table 2 and Table 3). In contrast, no RB mutants of PepSMV, PTV, or EcRV could be obtained, probably because these viruses induced severe necrosis in scions and killed plants too early.

### 3.4. Substitution K_472_E in PVY NIb (RdRp) Determines Adaptation to PM949 Resistance

Due to the similarity of phenotypes observed in PM949 and W4 (carrying *Pvr4*) upon inoculation with potyviruses, we tested whether the RB mutants selected by W4 could also break PM949 resistance, and vice versa, by mechanical inoculation (Table 3). The four W4-selected SON41p-G1 to SON41p-G4 PVY populations carrying the K_472_E codon substitution in NIb cistron were able to infect PM949 (20 of 20 inoculated plants). In contrast, the SON41p-G5 and SON41p-G6 populations that did not carry the K_472_E substitution were unable to infect PM949 (no infected plants out of 20 inoculated). This result suggests that the K_472_E substitution in NIb may be involved in the breakdown of PM949 resistance in addition to *Pvr4*. This was verified using a mutagenized cDNA clone of PVY-SON41p containing the K_472_E substitution (Janzac et al. 2010 [11]). Virus derived from this mutant clone was inoculated into 40 PM949 plants and infected all plants at the systemic level, 15 dpi. No additional nucleotide substitutions were observed in the NIb cistron of viral progeny in the four PM949 plants analyzed.

These results reveal that the NIb cistron determines the breakdown of PM949 resistance and that the K_472_E substitution is sufficient for breakdown of both *Pvr4* and PM949 resistance. Thus, the K_472_E substitution is one of the few examples of cross-infectivity effects, where a single mutation in the parasite results in the simultaneous breakdown of two or more resistances [12]. The fact that a single mutation in the PVY genome allows the simultaneous breakdown of a dominant and a recessive resistance may seem surprising given the contrasting modes of action usually considered for these two categories of resistance. Indeed, recessive resistance generally corresponds to a loss-of-function mode of action, where a host factor cannot be exploited by the pathogen, for example, due to of a lack of interaction with a pathogen protein [19]. In contrast, dominant resistance is usually based on the induction of defense responses following the recognition, whether direct or not, of a pathogen factor by a host resistance factor [21]. The dual effect of the PVY K_472_E mutation in resistance breakdown may be coincidental, with the same NIb domain being involved in physical interactions with two different plant ligands involved in either a recessive or in a dominant form of resistance. Alternatively, a resistance gene or different members of the same gene family may behave as dominant or recessive depending on their allelic forms or the alleles present on the other paralogue genes, as has been shown in the case of the eIF4E family [22,23]. A third hypothesis could be based on a gene-for-gene resistance mechanism as described by the guard model [24]. According to this model, a dominant monogenic resistance is expressed when triggered by the interaction (or alteration) of a host protein, the guardee, with (by) the pathogen’s avirulence factor, here NIb. Consistent with this model, resistance could be triggered in a dominant manner in some plant genotypes and could be determined by the absence or modification of a guardee protein in a recessive manner in other plant genotypes. In this framework, resistance breakdown could occur through a modification of NIb that would (i) allow its interaction with an alternative plant susceptibility factor in the context of the recessive resistance and simultaneously (ii) affect its interaction with the host guardee protein and thereby abolish the triggering of the resistance in the context of the dominant resistance.

### 3.5. Additional Amino Acid Substitutions in PVY NIb Are Associated with Adaptation to PM949 Resistance

The NIb cistron of all ten PM949-selected PVY populations from the graft-inoculation experiments was sequenced. In each PVY population, a single non-synonymous substitution was observed, with seven different substitutions in total (Table 2 and Table 3). These seven substitutions are distributed in three areas of the NIb cistron: At codon 47, from codon 94 to codon 108, and from codon 488 to codon 508. The latter area is close to codon position 472, that has been implicated in the breakdown of *Pvr4* and PM949 resistance. Several arguments suggest the involvement of these substitutions in the breakdown of PM949 resistance: (i) the previous finding that the K_472_E substitution in NIb is sufficient for breakdown of PM949 resistance, (ii) the fact that all PM949 RB mutants carry a single nonsynonymous mutation in the NIb cistron, (iii) the fact that several of these mutations occurred repeatedly and independently in PM949, such as the E_47_K (2 times) and M_502_I (3 times) mutations, and (iv) the fact that several of the mutations are located in the same NIb region or even in the same codon (mutations at positions 472, 488, 502; mutations E_508_G and E_508_G). Of the 10 PVY-SON41p-derived populations breaking PM949 resistance, only 2, carrying either the R_488_G or E_47_K substitution, were able to infect only 1 of the 20 inoculated W4 plants (Table 3). The first PVY population could not be analyzed further because it lost its infectivity during storage. A sample of the second PVY population collected from the infected W4 plant was able to infect 25 of 25 back inoculated W4 plants, showing a gain of pathogenicity, and the additional Y_271_C substitution was observed in the NIb cistron (Table 3). Therefore, PM949 served as a springboard to PVY-SON41p to break the *Pvr4* resistance, as has been shown for eIF4E-mediated recessive resistances in several solanaceous crops [12]. Indeed, upon mechanical inoculation with PVY-SON41p, no infection was observed in routine tests performed on >1500 W4 plants (Table 2). The frequency of infection of W4 by PVY populations from PM949 was significantly higher (0/1500 vs. 2/200 infections; *p*-value = 0.014; Khi^2^ test) (Table 3). Most of the RB-associated mutations (seven out of eight, with the exception of the M_502_I substitution) in PVY affect the local electrostatic potential or polarity of the NIb surface, and several of them (three of the five substitutions for which the NIb structure could be modelled) also affect the NIb structure (Table 3; Figure 2). Therefore, these mutations may be involved in RB by altering the interaction of PVY NIb with as yet unknown ligands, such as the plant proteins PABP, heat shock cognate 70-3, or the translation elongation factor eEF1A, which have been identified as interactors of potyvirus RdRp both *in vitro* and *in vivo* [25,26].They could also modulate NIb-RNA association, as RdRp is associated with the primer-template RNA duplex during replication.

### 3.6. Insights into the Durability Potential of Pvr4 and PM949 Resistance

Interestingly, *Pvr4* appears to have both higher durability potential to PVY-SON41p and a broader spectrum of action than PM949. Indeed, a significantly higher frequency of graft-inoculated plants generated RB mutants with PM949 (10 out of 10 grafts) than with W4 (6 out of 25 grafts; *p*-value = 4.4 × 10^−5^; Khi^2^ test). In addition, *Pvr4* confers resistance to two or three additional potyviruses compared with PM949 (Table 1). This reinforces the hypothesis that its spectrum of action may be a predictor of the potential durability of plant resistance, as observed for *pvr2* alleles in *C. annuum* [3]. Indeed, a resistance with a broad spectrum of action, thus effective against pathogen species separated by large evolutionary time scales, is more likely to be durable against pathogen populations separated by shorter evolutionary time scales [3].

In practice, PM949 resistance appears to be less attractive than *Pvr4* in terms of spectrum of action, durability potential and ease of introgression into elite cultivars. Worse, PM949 could act as an evolutionary springboard for PVY, facilitating the breakdown of *Pvr4* in a second step (Table 3). Therefore, *Pvr4* is a preferable source of resistance, and the use of PM949 resistance is not recommended, at least until *Pvr4* is broken down. Based on our results, the only value of PM949 would be to counter-select *Pvr4*-breaking PVY isolates lacking the NIb K_472_E substitution, should such isolates emerge.

## Figures and Tables

**Figure 1 viruses-15-01081-f001:**
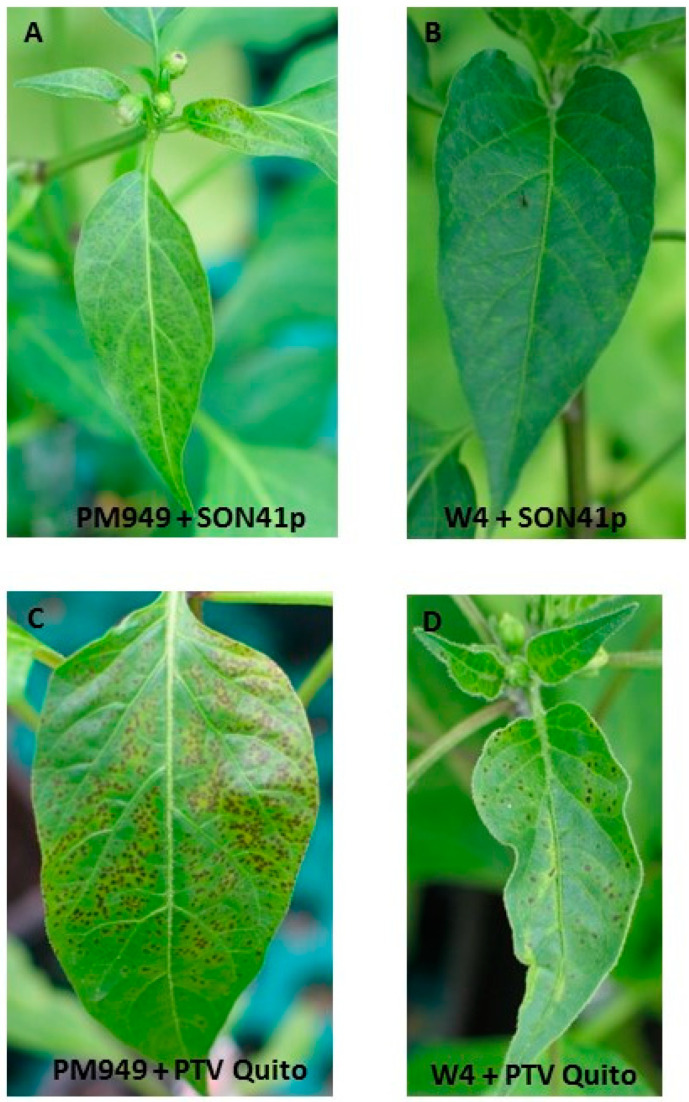
Hypersensitive-like reactions in graft-inoculated PM949 (**A**,**C**) and W4 (**B**,**D**) plants. PM949 and W4 scions were grafted onto potyvirus susceptible Yolo Wonder rootstocks, and the latter was inoculated one week after grafting by PVY-SON41p (**A**,**B**) or PTV isolate Quito (**C**,**D**).

**Figure 2 viruses-15-01081-f002:**
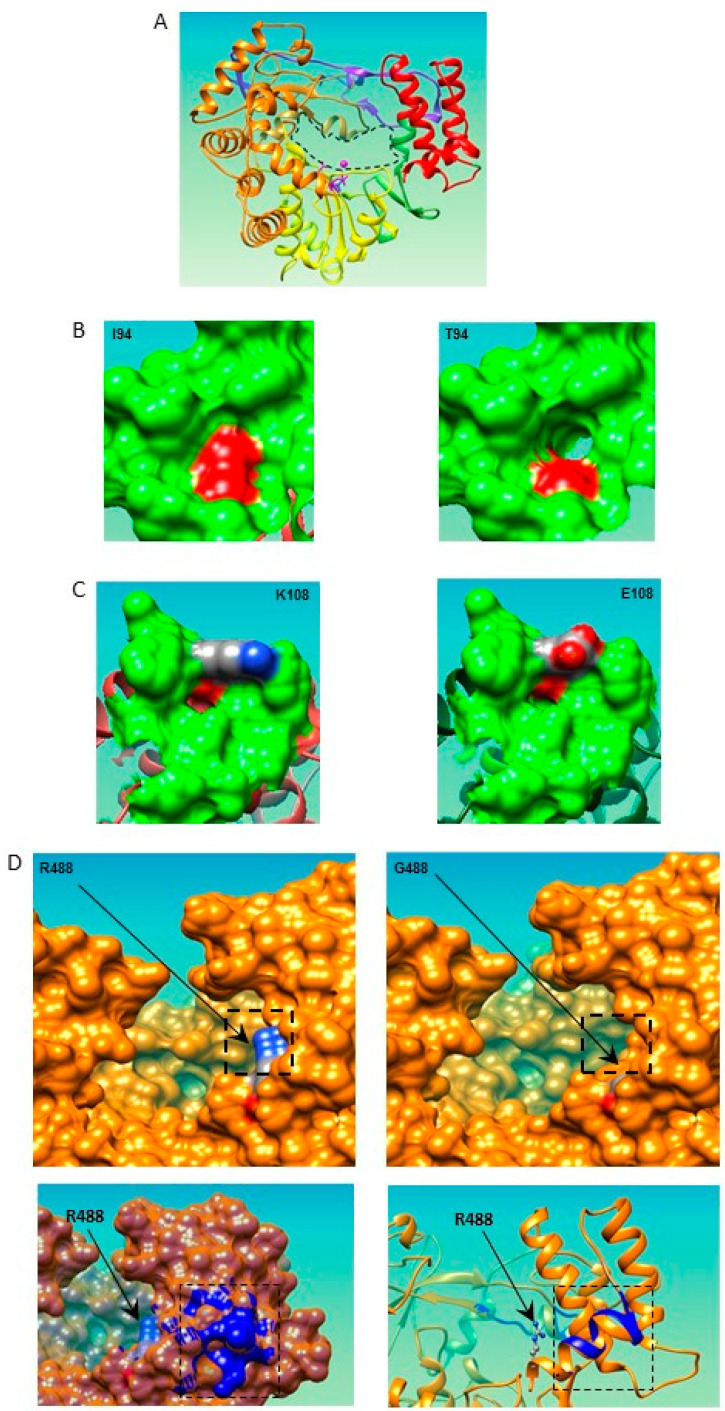
Structural models of the NIb (Nuclear inclusion b protein), i.e., the RNA-dependent RNA polymerase (RdRp) of PVY-SON41p and its mutants based on the structure of the RdRp of *Rabbit hemorrhagic disease virus* (RHDV) (Protein Data Bank 1KHV). The NIb of the eight different single-amino-acid mutants of the SON41p isolate breaking *Pvr4* and/or PM949 resistance (Table 2) was modeled using the Phyre2 protein modeling web portal [27] (http://www.sbg.bio.ic.ac.uk/~phyre2/html/page.cgi?id=index; accessed on 14 March 2015). For all these mutants, the highest scoring model was that of the RdRp of *Rabbit hemorrhagic disease virus* (RHDV; genus *Lagovirus*, family *Caliciviridae*) (Protein Data Bank 1KHV [28]), with at least 80% of the sequence modeled with 100% confidence. No satisfactory model could be obtained for the M_502_I, E_508_G, and E_508_K mutants of SON41p due to excessive coordinate uncertainties of the peptide backbone folding beyond residue 489. The other mutations associated with PM949 resistance breakdown that could be mapped were located in two distant regions of the protein, the finger and thumb domains (Table 3). (**A**) Global structural model of NIb from the PVY-SON41p isolate. The RHDV RdRp adopts the overall structure of a right hand with fingers, palm, and thumb domains, a folding shared by most other polynucleotide polymerases [29]. The finger, palm, and thumb domains are in orange, yellow, and red, respectively. The first 52 N-terminal amino acids were not modeled satisfactorily. The 63 N-terminal amino acids of the RHDV RdRp used as a model are shown in purple. In the crystal structure, this domain connects the finger and thumb domains. The GDD amino acid residues and the Mn^2+^ ion positioned in the catalytic site are shown in pink. The entrance of the RNA channel is delineated by a dashed black line. (**B**) Comparison of the local structure of PVY-SON41p and the SON41p-I_94_T mutant. In PVY-SON41p (left), the side chain of isoleucine 94 forms the bottom of a small crevice that becomes deeper upon substitution with the threonine residue (right), most likely due to the shorter side chain of threonine. Due to the hydroxyl moiety of threonine, the crevice also becomes more polar. (**C**) Comparison of the local structure of PVY-SON41p and the SON41p-K_108_E mutant. In PVY-SON41p (left), the lysine residue forms a straight, regular edge above a 15 Å-wide crevice. Upon substitution with glutamic acid (right), the shape of this edge is altered, with the two oxygen atoms (in red) of the side chain protruding to the outside. This substitution is accompanied by a net electrostatic change over the entire crevice from a slightly positive to a strongly negative state at physiological pH (not shown). (**D**) Comparison of the local structure of PVY-SON41p and the SON41p-R_488_G mutant. In PVY-SON41p (left), the nitrogen atom (blue sphere) of the arginine 488 side chain protrudes slightly 7 Å forward from the surface of the edge of the entrance of the RNA channel (enzyme thumb domain). At physiological pH, arginine 488 creates a strong positive surface potential. Upon substitution with glycine (right), a small depression 5 Å deep replaces the protruding lysine side chain, and the potential becomes neutral. This substitution could mitigate the likely strong electrostatic interaction between the negatively charged RNA backbone and arginine 488. This positive surface density is retained in the RHDV template. The glycine substitution also induces a more distant alteration of the first helix turn in the α-helix extending from amino acid 477 to 489. The structures of PVY-SON41p and its R_488_G mutant have been superimposed (lower panel; PVY-SON41p in orange and the R_488_G mutant in blue). Alterations are visible in both the backbone folding (lower right panel) and the surface topology (lower left panel).

**Table 1 viruses-15-01081-t001:** Behavior of PM949 and W4 pepper lines after mechanical inoculation with potyviruses.

Virus Species and/or Isolate *^a^*	Accession Number	Phylogenetic Cluster	Infectivity and Symptoms in PM949 *^b^*	Infectivity and Symptoms in W4 *^b,c^*
PVY-SON41p	AJ439544	PVY	Ø/Ø *^d^*	Ø/Ø
PVY-LYE72	EU334782	PVY	Ø/Ø	NEC/Ø
PVY-Chile1	FJ951646, FJ951645	PVY	NEC/Ø	NEC/Ø
PepSMV	X66027	PVY	NEC/Ø	NEC/Ø
PepYMV	AB541985	PVY	MO/MO	NEC/Ø
PTV-Quito	-	PVY	NEC/NEC	NEC/Ø
PTV-Cuzqueño2	EU495235	PVY	NEC/NEC	NEC/Ø
PepMoV-Texas	-	PVY	MO/MO	NEC/Ø
EcRV	EU495234	PVY	NEC/Ø	NEC/Ø
PVMV-Ivory Coast	DQ009807, MG334358	PVMV	MO/MO	MO/MO
ChiVMV-Taiwan	-	PVMV	MO/MO	MO/MO
TEV-HAT	M11458	TEV	MO/MO	MO/MO

*^a^* see main text for virus acronyms. *^b^* results from 20 plants per virus (2 independent experiments with 10 plants per virus). *^c^* after Janzac et al. 2009 [14]. *^d^* observations in inoculated cotyledons and leaves (left) and in apical uninoculated leaves (right). Ø: no symptoms and no virus detection in ELISA; NEC: necrotic lesions and virus detection in ELISA; MO: mosaic symptoms and virus detection in ELISA.

**Table 2 viruses-15-01081-t002:** Pathogenicity of PVY-SON41p towards pepper genotypes and mutations observed in the NIb cistron.

Pepper Genotype	Test Conditions *^a^*	Resistance Test (Infected/ Inoculated) *^b^*	Nonsynonymous Substitutions in the NIb Cistron
YW	Mechanical, 1 mpi	20/20	no
W4	Mechanical, 1 mpi	0/1500 *^c^*	-
PM949	Mechanical, 1 mpi	0/20	-
W4	Grafting, 5 mpi	6/25	SON41p-G1: D_88_N + L_234_H + K_472_E *^d^* SON41p-G2: E_390_G + K_472_E *^d^* SON41p-G3 and SON41p-G4: K_472_E *^d^* SON41p-G5: I_94_T + V_416_A + A_504_V SON41p-G6: M_365_I + G_471_E + S_478_N
PM949	Grafting, 5 mpi	10/10	E_47_K (2 plants) I_94_T (1 plant) K_108_E (1 plant) R_488_G (1 plant) M_502_I (3 plants) E_508_G (1 plant) E_508_K (1 plant)

*^a^* Inoculation method and duration of the resistance test in months post inoculation (mpi). *^b^* Infections based on PVY detection in apical uninoculated leaves by DAS-ELISA. *^c^* W4 is used in routine tests for cultivar registration on the French catalogue. *^d^* after Janzac et al. [11,14].

**Table 3 viruses-15-01081-t003:** Pathogenicity of PVY-SON41p NIb mutants towards *C. annuum* W4 and PM949.

PVY cDNA Clone or Isolate	Location and Impact of Mutation on NIb	Plant Genotype	Resistance Test (Infected/Inoculated) *^a^*	Mutations in the NIb of PVY Progeny *^b^*
SON41p-G1 to -SON41p-G4 *^c^*		W4	20/20 each isolate	no
		PM949	20/20 each isolate	no
SON41p-G5 *^d^*		W4	20/20	no
		PM949	0/20	
SON41p-G6 *^d^*		W4	20/20	no
		PM949	0/20	
SON41p-K_472_E (cDNA clone)	Thumb domain; change of surface charge	W4	50/50	no
		PM949	40/40	no
SON41p-E_47_K (2 isolates)	Bridge between thumb and finger domains; change of surface charge	W4	0/20 and 1/20 for the two isolates	Y_271_C *^e^*
		PM949	20/20 each isolate	no
SON41p-I_94_T	Finger domain; change of local structure and more polar	W4	0/20	
		PM949	20/20	no
SON41p-K_108_E	Finger domain; change of surface charge and local structure	W4	0/20	
		PM949	20/20	no
SON41p-R_488_G	Thumb domain; change of surface charge and local structure	W4	1/20	not determined
		PM949	20/20	no
SON41p-M_502_I (3 isolates)	Structure not modelled	W4	0/20 each isolate	
		PM949	20/20 each isolate	no
SON41p-E_508_G	Change of charge; structure not modelled	W4	0/20	
		PM949	20/20	no
SON41p-E_508_K	Change of charge; structure not modelled	W4	0/20	
		PM949	20/20	no

*^a^* Infections based on PVY detection in apical uninoculated leaves by DAS-ELISA at one mpi. *^b^* The NIb cistron of PVY population in two infected plants per virus isolate–plant genotype combination (when available) was sequenced. *^c^* the four isolates share the K_472_E nonsynonymous substitution in the NIb cistron (Table 2 in Janzac et al. [11]). *^d^* SON41p-G5 carries nonsynonymous substitutions I_94_T + V_416_A + A_504_V and SON41p-G6 carries nonsynonymous substitutions M_365_I + G_471_E + S_478_N in the NIb cistron (Table 2). *^e^* the isolate obtained from the infected W4 plant and containing the Y_271_C nonsynonymous substitution in the NIb cistron was used for back-inoculation to W4 seedlings, showing 25 infected plants of 25 inoculated and no additional mutation in the NIb cistron.

## Data Availability

All sequence data generated in this study are described in the present article.

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
