# Peer review of "A Single Nonsynonymous Substitution in the RNA-Dependent RNA Polymerase of Potato virus Y Allows the Simultaneous Breakdown of Two Different Forms of Antiviral Resistance in Capsicum annuum"

_viruses, 2023, doi:10.3390/v15051081_

Round 1

Reviewer 1 Report

IIn article of Moury et al. describe a new potyvirus resistance source in the Guatemalan accession C. annuum cv. PM949. PM949 is resistant to members of at least three potyvirus species, a subset of those controlled by Pvr 4.  The resistance progeny determined by two unlinked recessive genes conferring independently PVY resistance.  Comparison of Pvr 4 and PM949 resistances, sharing the same target in PVE, provides interesting insights in determinants of resistance durability. The experiments in this paper are well designed, performed and nicely presented. The results and findings from this paper are also solid and should be of importance to the field and providing more understanding of host gene interaction with viruses. I recommend the publication of this paper. I have no comments.

Author Response

Thank you for your review.

Reviewer 2 Report

Authors Benoît Moury and coworkers presented here manuscript entitled "A single nonsynonymous substitution in Potato virus Y RNA-dependent RNA polymerase permits simultaneously breakdown of two different forms of antiviral resistance in Capsicum annuum".

Authors characterised the source of resistance to PVY and two other potyviruses, present in Capsicum annuum cv. PM949. The segregation pattern in the F2 progeny indicates probable resistance determination by two independent recessive genes.

For comparison, resistance gene Pvr4 was used. It is conferring resistance to six potyviruses, including also PVY.

PVY mutation E472K, located in the NIb cistron, is breaking the resistance conferred by both Pvr4 pepper gene (dominant gene) and PM949 pepper cultivar (recessive gene).

Authors discuss several possible theories to understand this observation.

Other mutations also appeared during the work with different effects on the two resitance genes.

Finally, authors do not recommend to use the PM949 resistance, as it may facilitate resistance breaking of Pvr4 gene.

The manuscript is partially difficult to follow, more figures and graphs may enhance its readability and understandability.

English needs a minor revision, several phrases are unclear and the use of articles is not always correct.

Author Response

Thank you for your review.

Without further information it is a bit difficult to see which additional figures or graphs would be necessary. We decided to move the two figures that appeared as Supplementary documents in the main text. Moreover, we do not know if the reviewers had access to the Graphical abstract but this Graphical abstract summarizes the main findings of the article.

Concerning the revision of the English, we made an extensive revision of the whole article with quite a lot of modifications that we think have improved the quality of the English. You can check these revisions in the "change track" document which is joined.

Best regards,

Benoit Moury on behalf of co-authors
